# Environmental Conditions along Tuna Larval Dispersion: Insights on the Spawning Habitat and Impact on Their Development Stages

Stefania Russo [1,2,*] , Marco Torri [1,*] , Bernardo Patti [3] , Marianna Musco [1], Tiziana Masullo [1], Marilena Vita Di Natale [1,2], Gianluca Sarà [2] and Angela Cuttitta [1]

1   CNR-ISMed, National Research Council, Institute for Studies on the Mediterranean, 90145 Palermo, Italy; marianna.musco@ismed.cnr.it (M.M.); tiziana.masullo@ismed.cnr.it (T.M.); marilena.dinatale@ismed.cnr.it (M.V.D.N.); angela.cuttitta@ismed.cnr.it (A.C.)
2   DiSTeM-UNIPA, Department of Earth and Marine Science, University of Palermo, 90128 Palermo, Italy; gianluca.sara@unipa.it
3   CNR-IAS, National Research Council of Italy, Institute for the Anthropic Impacts and Sustainability in Marine Environment, 90149 Palermo, Italy; bernardo.patti@ias.cnr.it
*   Correspondence: stefania.russo@ismed.cnr.it (S.R.); marco.torri@ismed.cnr.it (M.T.); Tel.: +39-3293155216 (S.R.); +39-3299598064 (M.T.)

**Abstract:** Estimated larval backward trajectories of three Tuna species, namely, Atlantic Bluefin Tuna (*Thunnus thynnus*, Linnaeus, 1758), Bullet Tuna (*Auxis Rochei*, Risso, 1801) and Albacore Tuna (*Thunnus alalunga*, Bonnaterre, 1788) in the central Mediterranean Sea, were used to characterize their spawning habitats, and to assess the impact of changes due to the major environmental parameters (i.e., sea surface temperature and chlorophyll-a concentration) on larval development during their advection by surface currents. We assumed that the environmental variability experienced by larvae along their paths may have influenced their development, also affecting their survival. Our results showed that the Tuna larvae underwent an accelerated growth in favorable environmental conditions, impacting on the notochord development. In addition, further updated information on spawning and larval retention habitats of Atlantic Bluefin Tuna, Bullet and Albacore Tunas in the central Mediterranean Sea were delivered.

**Keywords:** *Thunnus thynnus*; *Auxis rochei*; *Thunnus alalunga*; ichthyoplankton; Mediterranean Sea; backward trajectories; Lagrangian simulations; spawning habitat; larval habitat; tuna

## 1. Introduction

The biotic and abiotic conditions of the water column, particularly the surface layers, can strongly influence the distribution and the abundance of the fish larval stages, thereby affecting the reproductive success of many fish species [1].

After the spawning events, the fate of eggs and larvae is uncertain. The high mortality rates of these crucial early life phases have a strong impact on recruitment success [2–4]. Marine currents carry and disperse pelagic eggs and larvae through different habitats [5], and consequently affect their fate, which is also driven by variability of environmental forcings and biotic interactions, manifesting deterministic chaos [6]. For instance, predation rather than physical, chemical and trophic properties of water masses are some examples [7]. Any environmental forcing and interactions eggs and larvae experiment along their path can affect the final balance of reproductive success of each individual spawning event. In addition, the influence of environmental factors may vary according to the ontogenetic stage [8–10]. Thus, the pelagic environment is so variable and maximization of the reproductive success is so unpredictable that to increase our understanding of how the resulting stochastic variability can influence the fate of pelagic fishes, such as anchovies, mackerel and tunas, is crucial. This large unpredictability is expected to affect

vastly vagrant species, such as adult Tunas, that migrate from feeding habitat to spawning grounds at long distances [11]. Indeed, while adults can select the locations to maximize the reproductive event's success, stochastic events are more effective in addressing the recruitment-related processes. In this, to limit the effects of unpredictability, natal homing behavior is beneficial and is considered a strategy used by many organisms, including Tunas [12,13], to increase the likelihood of reproductive success and maximize the larval survival. Tunas follow specific signals to commence reproductive events together, but this depends on species [14–17]. The spawning behavior of Bluefin Tuna, in particular, can respond to environmental or physical signs, or a mix of both [18]. In any case, the typical strategy is to ensure the survival of as many larvae as possible.

Despite the extent of larval life being relatively short compared with the organism's lifetime, the fate of early life stages is decisive for the future adult stock [19–21]. In fact, larval abundances are currently used as a proxy for assessing spawning stock biomass [22]. During this phase, faster-growing individuals are favored over slower-growing individuals, the latter being exposed for a longer time to a vulnerable status, characterized by higher mortality rates induced by predation and harder feeding conditions [2]. This time interval is also very short for Tuna species compared with other marine fish larvae [19].

During the early life phase, fish larvae can make only small individual, mainly vertical, movements, but they undergo passive travel because they cannot decisively oppose the currents [19,23,24]. Even if limited in extension, these horizontal movements are essential because larvae can be removed from nursery areas and advected towards more, or not more, favorable, retention areas [6]. The flexion stage of the notochord is a milestone because larval swimming and feeding abilities improve significantly [8–10]. They become able to hunt better, escape predators more effectively, and make their first active movements by escaping from passive current transport.

Tunas are top predators and targets of fisheries globally. They have an important ecological and economic role, influencing the structure and function of marine communities [25]. Often larval fish habitats are associated with specific oceanic regions with circulation systems generated by particular topographical features [26], and during the early stages, different tuna species overlap in these habitats globally [27]. In the Mediterranean Sea, Tunas' larval habitats seem to be linked to specific temperature conditions, hydrographic characteristics, oceanographic mesoscale structures such as gyres and fronts, and oligotrophic waters that ensure the larvae encounter fewer predators [28–30].

Three Tuna fish species reproducing in the Mediterranean Sea also overlap their spatial ranges during the vulnerable reproductive phase, i.e., Atlantic Bluefin Tuna (ABT) (*Thunnus thynnus*, Linnaeus, 1758), Bullet Tuna (*Auxis rochei*, Risso, 1810) and Albacore Tuna (*Thunnus alalunga*, Bonnaterre, 1788). Their life strategies are complex and different, but they have certain key traits in common, including spawning areas and the larval habitats. Fluctuations in their stock biomass also depend on the planktonic larval ecology, influencing mortality rates and reproductive success. The geographical distribution of Tuna species is closely linked to oceanographic conditions, which also influence their spawning behavior [18]. However, how environmental conditions affect larval survival and development during larval dispersal is still an open question.

Some authors showed that the Strait of Sicily is a spawning area for Tuna species [30–33], whose hydrodynamic complexity, but limited spatial extent, can represent a useful natural laboratory to understand the impact of the oceanographic processes and conditions during the offspring dispersal phase on larval development and survival. Actually, the local surface circulation is dominated by the flow of the Modified Atlantic Waters (also known as Atlantic Ionian Stream (AIS) [34]).

Profiting on the planktonic nature of Tuna early life stages, this study analyzed the backward trajectories of larval specimens of the three Tuna species that overlap spatially during the summer season, sharing their spawning grounds and larval habitats [28,35].

In order to investigate the fate of the larval stages and their relation with environmental conditions, we used a Lagrangian simulation approach, which is able to estimate the larval

path [36] and also to identify, by backward calculation, the spawning areas. This technique has already been successfully applied to study larval stages and their relationships with environmental conditions [37,38].

Often larval studies only rely on the environmental conditions observed in the geographical sites where larvae were collected. Instead, in this study it was possible to assess the conditions experienced by sampled larval specimens, from their sampling sites backward to their presumed hatching location (as estimated by Lagrangian simulations).

Here, we verify the hypothesis that the environmental conditions experienced by fish larvae along the phase of passive transport can affect their developmental rates. In addition, the observed changes in environmental factors were related to the notochord development state.

## 2. Materials and Methods

### 2.1. Field Sampling

Ichthyoplankton samples were derived from oceanographic surveys carried out in the Strait of Sicily during the summer period (June to August) from 2010 to 2016 (Figure 1).

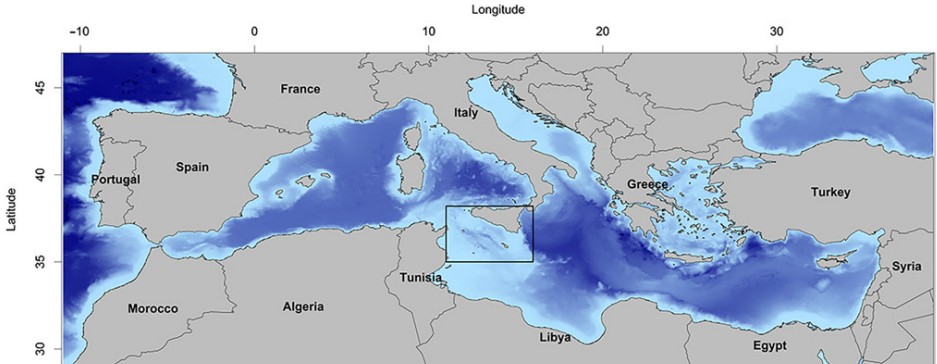

**Figure 1.** Study area.

The sampling was based on a regular grid of stations ($1/10° \times 1/10°$ in the continental shelf area, $1/5° \times 1/5°$ in the slope area) distributed along parallel transects perpendicular to the southern coast of Sicily.

Fish larvae were collected by oblique tows of a Bongo 40 plankton net and oblique or horizontal tows of a Bongo 90 plankton net, carried out at a ship speed of 2 knots. The sampling nets were towed from the surface to 100 m depth (Bongo 40) and in the surface mixed layer (Bongo 90).

### 2.2. Laboratory and Image Processing

After sampling, samples were immediately stored on-board in 70% ethanol, and subsequently processed in the land-based laboratory to identify Tuna larvae at the highest possible taxonomic level, according to Rodríguez et al. [4]. Images of each larva were acquired using stereomicroscopes with integrated cameras. The dedicated Image Pro Plus © (IPP) image management and processing software was used to obtain some morphometric parameters from the photos. The considered measurements derived from Catalán et al. [39].

### 2.3. Age Estimation

To assess the age (in days) of each individual Tuna larval specimen, we used the empirical relationships estimated in previous studies in the Mediterranean Sea on the daily increment of otolith rings related to standard length (SL) [40,41]. In particular, for Bullet Tuna, we used the relationship estimated by Laiz-Carrión et al. [41] for Mediterranean waters.

*2.4. Backward Trajectories Calculation*

Once larval ages of collected Tuna larval specimens were calculated, the corresponding larval backward trajectories (in terms of duration of simulation in days) were assessed, in order to infer the location of spawning grounds and the (satellite-based) environmental conditions that larvae experienced along their path, from the sampling sites backward towards the hatching sites.

Specifically, larval trajectories were simulated using General NOAA Oil Modelling Environment (GNOME), a software package designed by the NOAA Hazardous Materials Response Division [42]. Lagrangian elements (particles) movement is simulated within a geospatially mapped environment [43], which offers various opportunities for controlling input data to describe the transport of passive particles (in the present study, Tuna larvae) released at different sites [44,45]. In this study, the daily surface current field, i.e., the main "mover" of the fish larvae, used for the simulation runs, were from altimetry products, as distributed by the Copernicus Marine Service (CMEMS, http://marine.copernicus.eu/, accessed on 20 October 2021). In addition, horizontal diffusion was treated as a random walk process, calculated from a uniform distribution [46]. The GNOME default coefficient of $10^5$ cm$^2$ s$^{-1}$ was applied.

The influence of wind on surface circulation patterns was also taken into account, starting from a value-added 6-hourly gridded analysis of ocean surface winds [47]. Precisely, wind speed and directions were calculated from a zonal and meridional surface (10 m) and wind information was extracted using a 2.5 degrees of latitude × 2.5 degrees of longitude global grid for the geographical area 33°–40° N to 8°–20° E, as available in the link https://psl.noaa.gov/data/gridded/data.ncep.reanalysis.html, accessed on 20 October 2021 Extracted wind time series were included as additional external movers within GNOME backward Lagrangian simulations.

The start of the simulation runs was fixed at the sampling date of each collected larval specimen, for a duration in days corresponding to the estimated age of each larva based on its length, using the empirical relationships reported by García et al. [40] and Laiz-Carrión et al. [41]. In each simulation run, the release of particles was instantaneous from the selected sampling sites, corresponding to the geographical locations where at least one of the three fish species under study was recorded. The simulation consisted of three steps: (1) 1000 particles were positioned in each location of the selected sampling stations in the study area; (2) using GNOME, the direction and speed of the transport trajectory were calculated for the fixed duration of each simulation; and, (3) for each sampling station and each of the three species, the average final positions of released particles at the end of simulation runs, approximating the geographical location of the spawning sites, were evaluated and plotted.

The effect of the wind on the dispersal of particles was related to the expected vertical distribution of fish larvae in the water column. Given that the bulk of the larval abundance was likely to be concentrated from the surface down to the maximum depth of the mixed layer (from the analysis of temperature profiles using CTD probe, it was about 12 m on average over the six summer surveys included in our study), this reference depth layer was adopted for the simulations.

Taking into account the formulation reported in Patti et al. [36], the windage effect in Lagrangian simulations, i.e., the movement of particles induced by the wind, was set in terms of fractions of wind speed in the range 0.93–0.23%, with values corresponding, respectively, to depths of 1 m and 10 m.

Finally, we obtained probability clouds for the estimated backward trajectories of the released particles (see Supplementary Material, Figure S1h). The paths that impinged the coastline were excluded from the analyses. The calculation of centroids of the daily dispersion clouds made it possible to assess the backward larval path, to the hatching sites. The areas of these clouds represented the basis for the subsequent characterization of environmental conditions.

### 2.5. Remote Sensing Dataset

Satellite-based datasets were used to characterize the environmental conditions occurring in the identified areas. In particular, sea surface temperature (SST) and chlorophyll-a concentration (Chl-a) values were extracted for each point within the areas covered by each daily dispersion cloud. Subsequently, for each daily area, the average SST and Chl-a values were calculated. All satellite information was from the E.U. Copernicus Marine Service. In particular, we used daily data, with a spatial resolution of $0.01° \times 0.01°$ for SST [48], and of 1 km×1 km for Chl-a [49–53]. The reference geographical domain for SST and Chl-a data was, 34.5–39.0° N and 9–16.5° E during the spawning seasons (June-August) of years 2010 to 2016. This way, daily environmental conditions were associated with the paths of larval dispersion.

In order to investigate how larval development stages were affected by the environmental conditions experienced during their dispersion, the average SST and Chl-a values were obtained to include all simulation days in the calculation, i.e., from the catch day, backward to the estimated hatching day. In addition, in order to better characterize the spawning habitats of three Tuna species, bottom depth data extracted from the "Marpap package" [54] (ETOPO1 database) were associated with the final positions of backward larval trajectories.

### 2.6. Stages Classification

According to the development of the notochord, each larva was classified in preflexion, flexion and post-flexion stage, following the classification based on morphological characteristics of the notochord and caudal fin rays made by Blanco et al. [55].

We also divided each larval developmental stage into normal, early and late development, following the distribution of the SLs for each of the three Tuna species analyzed in this study.

The sub-categories were selected based on the quartiles of the frequency distributions of the three macro-categories (Figure 2, Table 1). Among the larvae evaluated in pre-flexion stage, normal pre-flexion was attributed for SL values below the 75th percentile, and late pre-flexion for SL values greater than or equal to the 75th percentile. For larvae evaluated in flexion stage, early flexion was attributed to larvae with an SL lower or equal to the 25th percentile, normal flexion for SL values between the 25th and the 75th percentile, and late flexion for SL values greater than or equal to the 75th percentile. Lastly, among larvae in post-flexion stage, early post-flexion was attributed for SL values below or equal to the 25th percentile, and normal post-flexion for values greater than the 25th percentile.

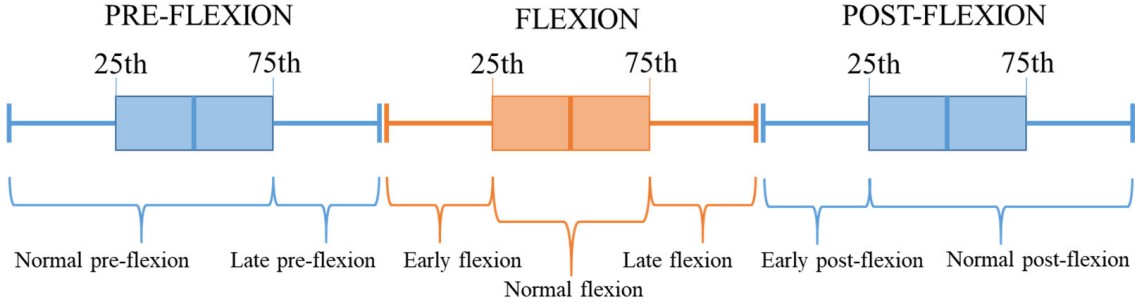

**Figure 2.** Classification of developmental stages into sub-categories ("Early", "Normal" and "Late").

The comparisons were accomplished, including all available data in the analysis (surveys 2010–2016), and on a year-by-year basis, when the number of observations was large enough.

**Table 1.** Classification of Tuna larvae in developmental categories and corresponding number of observations by species.

| Macro-Category | Sub-Category | ABT | Bullet | Albacore |
|---|---|---|---|---|
| Pre-flexion | Normal | 98 | 63 | 21 |
| | Late | 33 | 22 | 7 |
| Flexion | Early | 10 | 16 | 10 |
| | Normal | 18 | 31 | 20 |
| | Late | 10 | 16 | 10 |
| Post-flexion | Early | 9 | 8 | 2 |
| | Normal | 24 | 23 | 5 |

Non-parametric Kruskal–Wallis test was used to evaluate the significance of the differences in the medians of environmental parameters (SST and Chl-a) among the larval developmental stages. Pairwise comparisons were then carried out using the Wilcoxon rank-sum test in case of significant differences between groups ($p < 0.05$).

## 3. Results

### 3.1. Backward Trajectory Calculation

The estimated maximum larval age was 21 days, attributed to an ABT specimen sampled in the 2016 survey. Considering the whole available dataset, the observed average ages of collected larvae were 5 days for ABT, 8 days for Bullet Tuna, and 7 days for Albacore Tuna. The year with the lowest average age for ABT was 2010, due to the large number (60) of 1-day old specimens collected in the same sampling site, whereas the highest mean ages were recorded in surveys in 2014 and 2015. The estimated mean age of Bullet Tuna larvae was relatively stable over the sampling years, except in 2016, which was characterized by a low number of observations. Albacore Tuna larvae showed the lowest estimated average ages in all surveys.

The backward trajectories of some longer-lived larvae, by species and by year are reported in Figures S1–S3. It is worth noting the case of ABT and Bullet in 2014, where the trajectories mimic the average AIS path (Figures S1e and S2e). Pattern distribution of ABT larvae origins evidences their concentration in the southern part of the study area, off the southernmost tip of Sicily (Capo Passero) (Figure 3a). This applies also to many of the larvae whose estimated hatching site is in the western sector (Figure S1). Similar patterns are evident for the other two Tuna species, even though for Bullet Tuna the final positions of backward trajectories appear to be distributed in shallower areas (Figure 3b,c). In general, the paths of larvae of the same age can differ considerably. Surface currents can advect them at relatively long distances, while in some cases they can be trapped by local mesoscale oceanographic structures.

The variability in spawning sites seems to reflect the complex hydrodynamics of the study area. In particular, the Capo Passero area, characterized by a thermohaline front and warmer/saltier surface waters favoring the aggregation of larval stages of different fish species, including Tunas [30,56–58], confirms its features also by the analysis of backward trajectories. Generally, Tuna larvae that hatch in this area are trapped by the thermohaline front and by the local mesoscale circulation, which favor concentration processes also for larvae born elsewhere as long as they are advected there by the average path of surface currents (AIS). Larvae originating from spawning events in the Ionian Sea tend to also converge south of Capo Passero, where they join with larvae from the western sector (Figures S1–S3).

### 3.2. Characterization of Estimated Hatching Sites

The observed temperatures at origin points of estimated backward trajectories suggest different hatching temperatures for the three species (Table 2, Figure 4a,d,g). Specifically, the minimum temperature value is about 3 degrees higher for Albacore compared with ABT. However, the average temperature values are all in the range of 25–26 °C. In addition, while ABT larvae mainly occurred in waters with a surface temperature around 25–26 °C,

Bullet showed a wider thermal range skewed towards warmer waters. The same applies to Albacore, though results for this species is based on few observations.

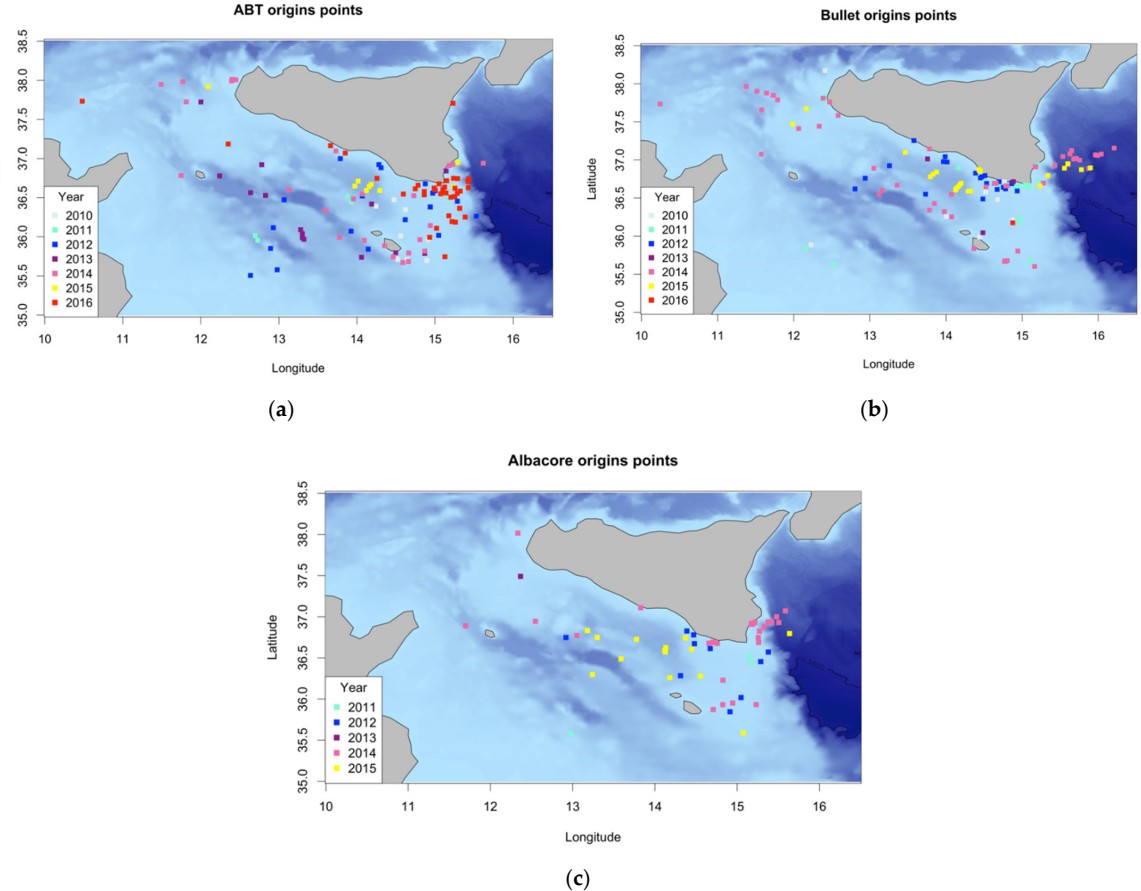

**Figure 3.** Calculated average origin point by year, for ABT (**a**), Bullet (**b**) and Albacore tuna (**c**).

Figure 4b,e,h shows that sampled larvae of the three species originated in highly oligotrophic waters, with quite low average concentration values (Table 2) especially for ABT.

As far bottom depth distribution is concerned, for ABT, the hatching site is more concentrated below −500 m. The median depth of Bullet (Table 2) also reflects the shallow water habitat of this species. For the same species, greater bottom depth values refer to origin points located in the Ionian Sea, and reflect the local orography, characterized by a narrow shelf area and a very steep continental slope but very close to the coastline.

**Table 2.** Environmental conditions, SST, Chl-a and bottom depth, in the estimated hatching sites for the three Tuna species.

| Environmental Variable | | ABT | Bullet | Albacore |
|---|---|---|---|---|
| SST | Minimum value | 20.47 | 20.71 | 22.82 |
| | Maximum value | 28.67 | 28.67 | 28.47 |
| | Median value | 25.16 | 25.98 | 26.32 |
| Chl-a | Minimum value | 0.029 | 0.033 | 0.029 |
| | Maximum value | 0.174 | 0.280 | 0.346 |
| | Median value | 0.048 | 0.055 | 0.049 |
| Bottom Depth | Minimum value | −1607 | −2470 | −2484 |
| | Maximum value | −15.7 | −9.00 | −11.28 |
| | Median value | −283 | −209 | −507 |

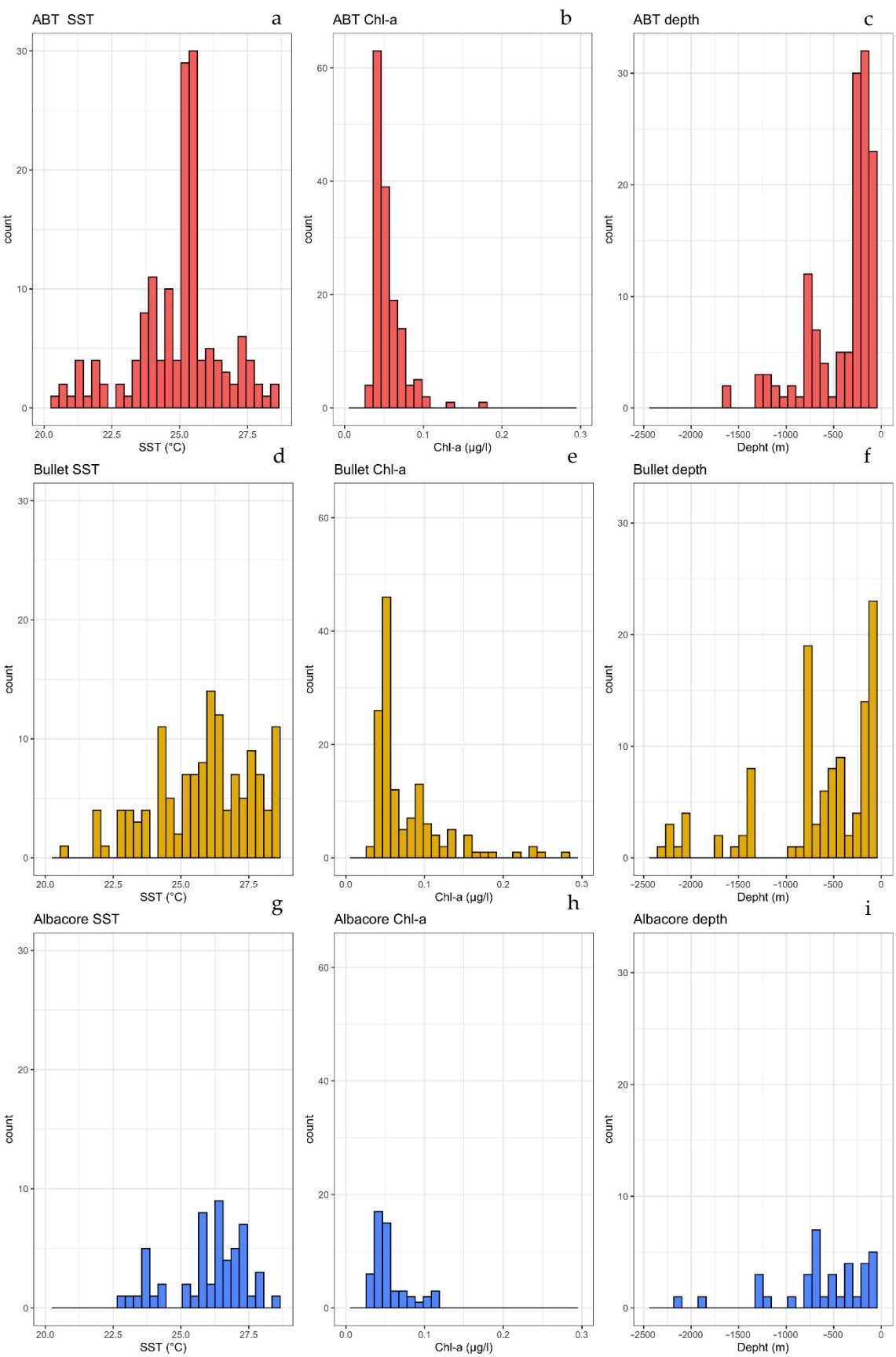

**Figure 4.** Histograms of environmental conditions, SST (**a**,**d**,**g**), Chl-a (**b**,**e**,**h**) and bottom depth (**c**,**f**,**i**), in the estimated hatching sites for the three Tuna species.

### 3.3. Relationships among Developmental Larval Traits and Environmental Forcings

Comparing the different developmental stage sub-groups (i.e., "Early", "Late" and "Normal") in relation to the impact of the environmental conditions experienced by larvae along their paths, some differences were found. In the case of ABT at the flexion stage, the Kruskal–Wallis test did not evidence any significant difference. On the contrary, for Chl-a, the difference between normal pre-flexion and late pre-flexion larvae was significant ($p < 0.001$; Figure 5a), with late pre-flexion associated with higher Chl-a concentrations.

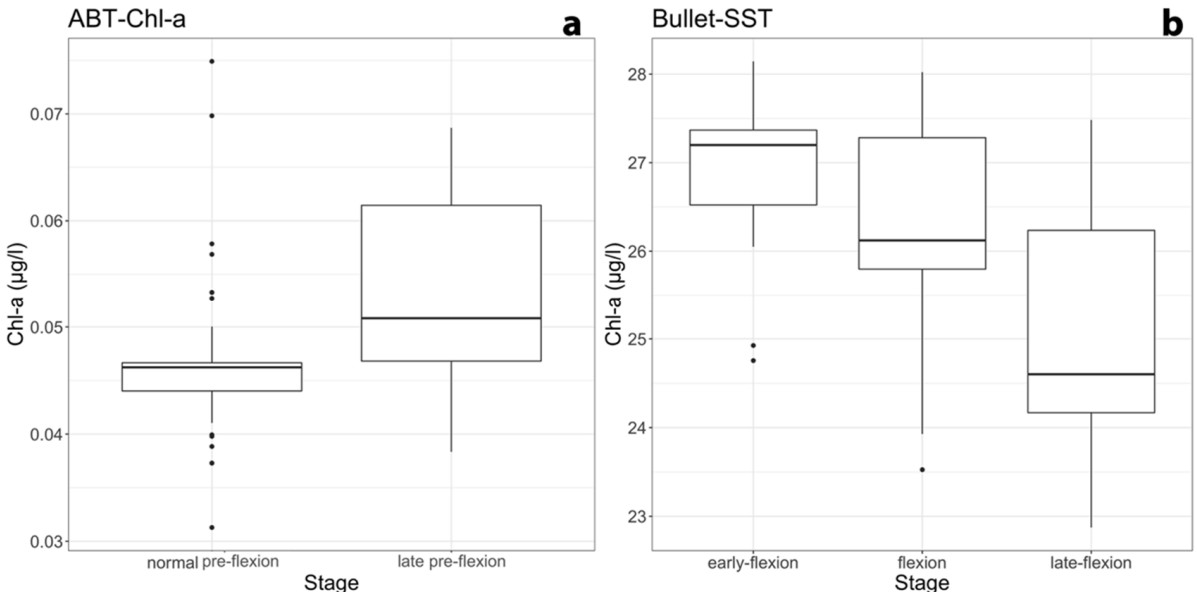

**Figure 5.** Box plots of developmental stage sub-groups in relation to environmental conditions experienced by Tuna larvae along their path, including information from all surveys. The two sub-plots show the cases where significant differences between sub-groups were found. (**a**) ABT vs. Chl-a and (**b**) Bullet vs. SST.

For Bullet larval specimens at the flexion stage, the Kruskal–Wallis test evidenced a significant overall difference in SST experienced along the path between sub-groups ($p < 0.01$; Figure 5b). Using pairwise comparisons, a significant difference was found between early and late development (Wilcoxon test, $p < 0.01$). Therefore, the early development larvae were found at higher temperatures and the late development larvae at lower temperatures.

No significant differences were found for Albacore across all years.

Examining ABT data by year, only in 2016 were some significant differences detected when comparing "normal pre-flexion" and "latepre-flexion" larval stages vs. SST ($p < 001$; Figure 6a) and vs. Chl-a ($p < 0.001$; Figure 6b), with late pre-flexion associated with lower SST and higher Chl-a concentrations.

No significant differences were found for Bullet or Albacore Tunas for the year-by-year analysis.

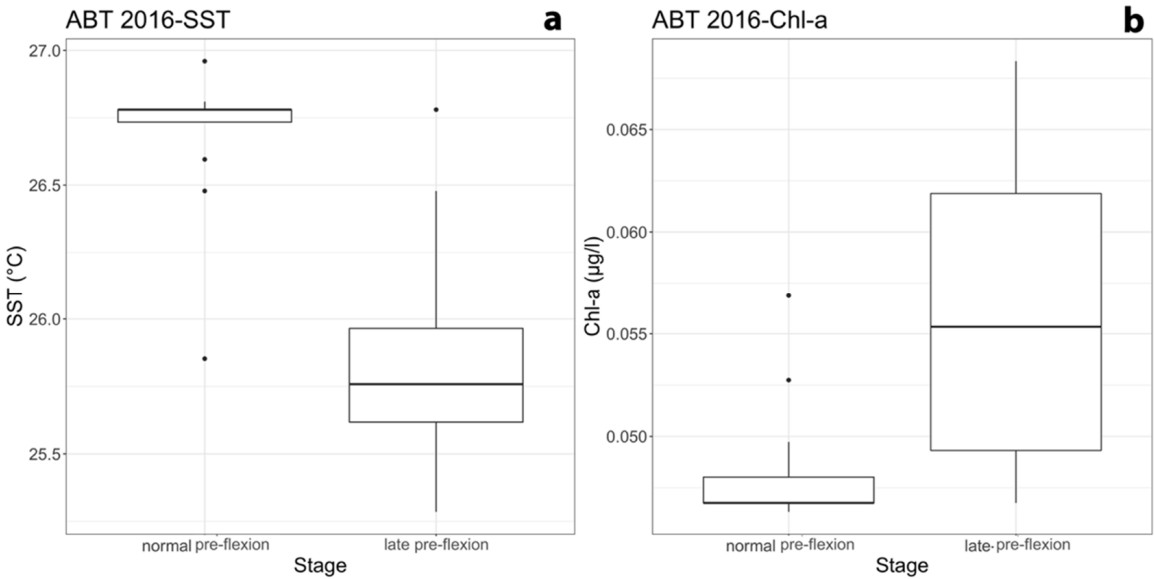

**Figure 6.** Box plots of developmental stage sub-groups in relation to environmental conditions experienced by Tuna larvae along their path (data from survey in 2016). The two sub-plots show the cases where significant differences between sub-groups were found. (**a**) ABT vs. SST and (**b**) ABT vs. Chl-a.

## 4. Discussion

The obtained results represent a further step forward the characterization of common spawning areas shared by three important Tuna species in the Strait of Sicily. Larval sampling was conducted during the spawning season, covering an area where larvae experience their development. Spatial distribution analysis of larvae highlighted different patterns among species. In addition, environmental variables seem to be helpful in describing the differences in the larval developmental stages.

The estimated ages allowed us to verify heterogeneity from year to year, indicating multiple spawning events within the studied area at different times. In fact, Tuna fish are multiple batch spawners, and the reproductive events follow one another within their reproductive season [59].

The analysis of backward trajectories showed the paths of each individual larva from the hatching to the catch. In agreement with other studies focused on the same region, AIS was one the main drivers affecting larval advection [36,60,61]. The surface current transported the planktonic fish stages and concentrated them in the southeastern retention area, where other larvae coming from the eastern sector also converge. Through backward Lagrangian simulations runs, the environmental conditions experienced by larvae throughout their lives were evaluated, starting from the origin points of their estimated backward trajectories, which represent the hatching sites. Considering the relatively short hatching time of the Tuna species (e.g., for ABT [62]), this study improved our knowledge about the spawning environment selected by the adults in the study area. The geographical positions of the origin points of backward larval trajectories varied significantly by species and by year. However, in all cases, the larval trajectories reflected the dominant path of the local surface current in the spawning and adjacent areas.

The larvae born at the edge of the thermohaline front tended to be trapped in this area. It was also interesting to note how larvae with similar age, depending on their origins, could experience different paths, covering large or short distances, and could often be found in common retention areas. This, of course, could also impact their development and/or survival. The events that affect larval development, along their path from the spawning sites, are fundamental in assessing reproductive success. In this framework, local oceanographic processes play a crucial role in determining the spatial distribution of

the planktonic fish stages, controlling the advection from the spawning areas to more or less suitable retention zones [63,64]. Other important factors affecting larval development and survival are the maternal effect and stochastic events during transport. The choice of the spawning grounds by adult specimens impact on the environmental conditions experienced by their offspring. Even the water masses' origin can affect larval growth, as shown by previous studies on Bullet Tuna [41,65,66].

The analysis of available satellite data (SST and Chl-a) and bathymetry associated with larval trajectories has increased our knowledge about the Tuna spawning habitat in the Strait of Sicily. Some observed patterns, such as the coastal attitude of Bullet or the different distributions depending on water temperature, correspond to what was verified by previous studies on the larval habitat [30]. In particular, backward trajectories analysis highlighted the different origins for larvae from the same sampling area, also delivering further insights into the maternal effect.

We hypothesized that the environmental conditions experienced by larvae along their passive drifting phase would have affected the development of these organisms. In fact, the individual's experience along the paths appears to influence the stage of the notochord, anticipating or delaying its development for ABT and Bullet. We hypothesized that warmer waters could anticipate development, and low Chl-a indicated unfavorable nutritional conditions, slowing the notochord development. Our results confirmed the hypotheses regarding temperatures, as early developing larvae were found in warmer waters, especially in the case of Bullet. For Chl-a, we found the opposite trend to what was hypothesized. This could be due to competition phenomena. We hypothesize that the occurrence of more competitors and predators in richer environments can drive lower feeding rates and energy waste due to escaping and hunting behavior, thereby leading to late notochord development. In this framework, warmer temperatures could lead Bullet, the fastest-growing species among those analyzed here [40], to a more accelerated development than the other Tunas. This, in turn, could lead to resource competition phenomena and/or early juvenile Bullet Tuna preying behavior on other Tuna species, as hypothesized by Bakun and Broad [67].

Condition, growth and survival in marine fish larvae are influenced by food [68], temperature [69], hydrographic patterns [70], and environment in general [71]. However, the parents' genotype could also be an essential factor [72,73]. In this study we did not evaluate any genotypic differences and a parental effect due to purely genetic factors. Embryo and larval characteristics, developmental rate, and metabolism are affected by the parents' genotype [72,73]. Together with the environmental parameters, they can lead to the manifestation of morphological differences and can affect the success of recruitment.

More in-depth studies involving genetic aspects are recommended, especially after recent tagging studies on adult ABT in the same reproductive area have shown different adult migratory behaviors [74]. However, the same study did not clarify whether these migratory patterns are related to two different subpopulations or whether the observed behavior is linked to different spawners' sizes. Therefore, the origin of the ABT population in the area is still an open question.

The early life stage analyzed here represents the most critical period in the life history of fish, affected by the highest mortality rates [75]. Understanding Tuna population dynamics is essential for determining the fundamental features of survival processes in the early life stages [76]. It is also necessary to understand the mechanisms affecting recruitment success for the three Tuna species analyzed here. In fact, understanding the links between ocean patterns, spatial distribution and paths of early life stages, and other environmental parameters is crucial for the sustainable management of fishery resources [77,78].

## 5. Conclusions

The Lagrangian simulation approach adopted in this study has improved our understanding of Tuna reproductive biology processes and early life history. The origin points

calculated varied significantly by species and by year, but the final position where the larvae were found reflect the surface current in the area.

We analyzed organisms that originated from multiple spawning events, finding that, regardless of the spawning area, the larvae released into the area ended up concentrating in a common retention area related to a local frontal mesoscale oceanographic feature. Therefore, the AIS seems to be fundamental in larval advection, and the front plays a key role trapping the larvae in a specific environment. The spawning habitat and the larval habitat have common characteristics, although some larvae have undertaken long transport routes.

We have shown that the environmental conditions experienced by the organisms, in accelerating or slowing down some relevant features of individual fish development (i.e., pre-flexion and flexion stages), could be fundamental for their survival from the first days of life. However, there is still a need for further insights into the physical processes affecting larval fate. Finally, our approach was intended to provide useful information to support Tuna fisheries management. Identifying spawning and retention areas for spawning products could be the baseline for developing fishery-independent recruitment indices. Furthermore, as suggested by Mariani et al. [23], better information on spawning areas and larval habitats can help establish marine protected areas or areas closed to fishing, for a valid protection strategy for these important fish species.

**Supplementary Materials:** The following supporting information can be downloaded at: https://www.mdpi.com/article/10.3390/w14101568/s1, Figure S1: ABT backward trajectories maps year-by-year, and an example of the cloud points derived from Lagrangian simulations; Figure S2: Bullet backward trajectories maps year-by-year; Figure S3: Albacore backward trajectories maps year-by-year.

**Author Contributions:** S.R., A.C. and G.S. contributed to the conception and design of the study. M.T., B.P., M.M., T.M., M.V.D.N. and A.C. conducted the sampling. S.R. and A.C. performed taxonomic identification. S.R. carried out the measurement and stages classification. S.R. organized the database. B.P. carried out the backward Lagrangian simulations. S.R. and M.T. performed the statistical analysis. S.R. wrote the first draft of the manuscript. G.S., A.C. and B.P. secured funding for data sampling and analysis. All authors have read and agreed to the published version of the manuscript.

**Funding:** Fondo Sociale Europeo Sicilia 2020 supported SR's PhD research presented in this study. Data collection was mainly supported by the Italian National Research Council (CNR) through the USPO office, and by the FAO Regional Project MedSudMed Assessment and Monitoring of Fishery Resources and Ecosystems in the Straits of Sicily, funded by the Italian Ministry MIPAAF and co-funded by the Directorate-General for Maritime Affairs and Fisheries of the European Commission (DG MARE). Sampling activity was also supported by other research projects, namely, Grandi Pelagici, funded by the Sicilian Regional Government, and SSD–PESCA and the Flagship Project RITMARE—The Italian Research for the Sea—coordinated by the Italian National Research Council and funded by the Italian Ministry of Economic Development and the Italian Ministry of Education, University and Research within the National Research Program 2011–2013.

**Informed Consent Statement:** Not applicable.

**Data Availability Statement:** The raw data supporting the conclusions of this article will be made available by the authors, without undue reservation.

**Acknowledgments:** We thank all the CNR technicians for their support during species identification, and Grazia Maria Armeri and Carlo Patti for their support during oceanographic surveys and sampling collection. We thank Emanuele Gentile, master of the R/V Urania and R/V Minerva Uno, and his crew, for their work supporting plankton sampling during oceanographic cruises.

**Conflicts of Interest:** The authors declare no conflict of interest.

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
