# Peer review of "Environmental Conditions along Tuna Larval Dispersion: Insights on the Spawning Habitat and Impact on Their Development Stages"

_water, doi:10.3390/w14101568_

Round 1

Reviewer 1 Report

Please limit self-citations to a maximum of 10% References (Marco Torri, Bernardo Patti, Angela Cuttitta). There are far too many self-citations and there are not enough references to the latest publications on similar topics. From 2020-2022, I found only 4 publications in References.

The chapter Conclusions needs to be supplemented and improved. The chapter Conclusions largely contains references to other publications, while it does not contain enough information on the importance of the analyzes carried out in the manuscript. 

Some excerpts from the chapter Conclusions seem too obvious, e.g.  "The fate of the offspring of pelagic fishes, subsequent to each one of their individual spawning events, is uncertain. During the first days, eggs and larvae are passively transported by the surface currents, so their fate is dictated by stochastic events and reproductive habitat selection car-ried out by the adults. Food, temperature, and hydrographic patterns affect growth and survival of fish larvae. In this framework, larval backward trajectories of three Tuna species, namely At-lantic Bluefin Tun."

Author Response

We thank the rev#1 for the comments. We accepted all suggestions. We attached a pdf that include a point-by-point response to the reviewer's comments

Reviewer 2 Report

 I think the work can be published. In my case, I accept the publication.

Author Response

We thank the reviewer for the comment.

Reviewer 3 Report

The paper "Environmental conditions along tuna larval dispersion: insights on the spawning habitat and impact on their development stages" by the authors Stefania Russo , Marco Torri , Bernardo Patti , Marianna Musco , Tiziana Masullo , Marilena Vita Di Natale , Gianluca Sarà and Angela Cuttitta is a very good manuscript.

The topic is of interest for the readers and ca be published without further improvement.

Author Response

We thank the reviewer for the comment.

Round 2

Reviewer 1 Report

The revised manuscript may be accept in present form.